# Novel Mycoviruses Discovered from a Metatranscriptomics Survey of the Phytopathogenic *Alternaria* Fungus

**DOI:** 10.3390/v14112552

**Published:** 2022-11-18

**Authors:** Wenqing Wang, Xianhong Wang, Chunyan Tu, Mengmeng Yang, Jun Xiang, Liping Wang, Ni Hong, Lifeng Zhai, Guoping Wang

**Affiliations:** 1Key Laboratory of Plant Pathology of Hubei Province, College of Plant Science and Technology, Huazhong Agricultural University, Wuhan 430070, China; 2Key Laboratory of Horticultural Crop (Fruit Trees) Biology and Germplasm Creation of the Ministry of Agriculture, Wuhan 430070, China; 3College of Life Science and Technology, Yangtze Normal University, Chongqing 408000, China

**Keywords:** mycovirus, *Alternaria*, metatranscriptomics, genome, positive- and negative single-stranded RNA viruses

## Abstract

*Alternaria* fungus can cause notable diseases in cereals, ornamental plants, vegetables, and fruits around the world. To date, an increasing number of mycoviruses have been accurately and successfully identified in this fungus. In this study, we discovered mycoviruses from 78 strains in 6 species of the genus *Alternaria*, which were collected from 10 pear production areas using high-throughput sequencing technology. Using the total RNA-seq, we detected the RNA-dependent RNA polymerase of 19 potential viruses and the coat protein of two potential viruses. We successfully confirmed these viruses using reverse transcription polymerase chain reaction with RNA as the template. We identified 12 mycoviruses that were positive-sense single-stranded RNA (+ssRNA) viruses, 5 double-strand RNA (dsRNA) viruses, and 4 negative single-stranded RNA (−ssRNA) viruses. In these viruses, five +ssRNA and four −ssRNA viruses were novel mycoviruses classified into diverse the families *Botourmiaviridae, Deltaflexivirus, Mymonaviridea, and Discoviridae*. We identified a novel −ssRNA mycovirus isolated from an *A*. *tenuissima* strain HB-15 as Alternaria tenuissima negative-stranded RNA virus 2 (AtNSRV2). Additionally, we characterized a novel +ssRNA mycovirus isolated from an *A*. *tenuissima* strain SC-8 as Alternaria tenuissima deltaflexivirus 1 (AtDFV1). According to phylogenetic and sequence analyses, we determined that AtNSRV2 was related to the viruses of the genus *Sclerotimonavirus* in the family *Mymonaviridae*. We also found that AtDFV1 was related to the virus family *Deltaflexivirus*. This study is the first to use total RNA sequencing to characterize viruses in *Alternaria* spp. These results expand the number of *Alternaria* viruses and demonstrate the diversity of these mycoviruses.

## 1. Introduction

Mycoviruses are wide distributions in the major taxonomic groups of filamentous fungi, yeasts, and oomycetes [1,2,3,4,5,6,7]. Most of the mycoviruses genomes are linear positive-sense single-stranded RNA (+ssRNA) or double-stranded RNA (dsRNA) [1,2,5,7]. Of these mycovirus genomes, a few have been found to include circular single-stranded DNA (ssDNA) or linear negative-sense single-stranded RNA (−ssRNA) [8,9,10,11,12,13,14,15,16].

*Alternaria* spp. can cause essential diseases in cereals, ornamental plants, vegetables, and fruits, and more than 95% of the species reported globally can become facultative parasites on different plants [17,18,19,20]. A recent study revealed that six *Alternaria* species are the causal agents of pear black spot diseases, resulting in massive economic losses in China [20].

To date, members of the genus *Alternaria* have been found to harbor several mycoviruses. Some *A*. *alternata* strains have been found to contain many uncharacterized virus-like dsRNAs [21,22]. Mycoviruses discovered in the strains of different species of the genus *Alternaria* have been classed into twelve families. Among these families, the family *Chrysoviridae* includes Alternaria alternata chrysovirus 1 (AaCV1) and Alternaria solani chrysovirus 1 (AsCV1) [23,24,25,26]. The family *Partitiviridae* includes Alternaria alternata partitivirus 1 (AaPV1) and Alternaria tenuissima partitivirus 1 (AtPV1) [27,28]. The family *Totiviridae* includes Alternaria arborescens victorivirus 1 (AaVV1) and Alternaria alternata victorivirus 1 (AalVV1) [29,30]. The genus *Botybirnavirus* includes Alternaria botybirnavirus 1 (ABRV1) and Alternaria alternata botybirnavirus 1 (AaBRV1) [31,32,33]. The family *Mitoviridae* includes two mitovirus called Alternaria arborescens mitovirus 1 (AaMV1) and Alternaria brassicicola mitovirus 1 (AbMV1) [34,35]. The family *Endornaviridae* includes Alternaria brassicicola endornavirus 1 (AbEV1) [36]. Alternaria alternata virus 1 (AaV1), which was identified in the proposed family *Alternaviridae*, was isolated from an *A*. *alternata* strain [37]. Importantly, AaV1 is the first dsRNA virus to be identified with both 5′ cap and 3′poly(A) structures on its genomic segments [38]. Moreover, strains of the genus *Alternaria* have been found to harbor Alternaria dianthicola dsRNA virus 1 (AdRV1), Alternaria longipes dsRNA virus 1 (AlRV1), and Alternaria alternata virus 1 (AaV1) which are unclassified dsRNA mycoviruses [37,38,39,40]. The member in the family *Hypoviridae* was found to include Alternaria alternata hypovirus 1 (AaHV1) [41]. In addition, Alternaria brassicicola fusarivirus 1 (AbFV1) and Alternaria solani fusarivirus 1 (AsFV1), which are positive ssRNA mycoviruses, are isolated in *A*. *brassicicola* and *A*. *solani*, respectively. AbFV1 and AsFV1 have been grouped with members of the recently proposed family *Fusariviridae* [42,43]. The other +ssRNA mycoviruses, Alternaria alternata magoulivirus 1 (AaMOV1) has been found to belong to the genus *Magoulvirus* in the family *Botourmiaviridae* [24]. Recently, full-length cDNA sequences of nine viruses were obtained by high-throughput sequencing in an *Alternaria dianthus* strain HNSZ-1 [44]. Among the nine viruses, five were confirmed to novel members in the families *Hypoviridae*, *Totiviridae*, *Mymonaviridae,* and a provisional family *Ambiguiviridae* [44]. Two −ssRNA, named Alternaria tenuissima negative-stranded RNA virus 1 (AtNSRV1) and Alternaria dianthus negative-stranded RNA virus 1 (AdNSRV1), belonged to the genus *Sclerotimonavirus* in the family *Mymonaviridea* [44,45]

In most cases, the host does not experience any phenotypic changes as a mycovirus infection [1,2,5,7]. Some mycoviruses, however, can cause debilitating symptoms in their hosts, including morphological changes, toxin production, and hypovirulence [4,5,6,7]. Among these hypovirulence-associated mycoviruses, some virus-infected strains can be used as biocontrol agents to prevent and treat fungal diseases in plants [3,7,45,46,47,48,49]. In some cases, phenotypic changes are result of mycoviruses harbored in the *Alternaria* fungi. For instance, AaCV1 not only restricted the growth of the host fungus but also rendered the host hypervirulent to the plant [26]. AaCV1-AT1 could reduce the growth rate and sporulation ability of the *A*. *tenuissima* strain [25]. AaV1 can cause impaired growth and unusual pigmentation in the host [38]. AaHV1 also can confer hypovirulence in other plant pathogenic fungi [41]. AdNSRV1 might be related to the phenotypic change of the host fungus [44].

In recent years, metatranscriptomics has been widely used in virus discovery. Many novel viruses have been discovered in fungi, which has significantly promoted the progress of viromics research and has sped up the discovery and understanding of unknown viruses [13,44,50,51,52,53,54,55,56,57,58,59]. In addition, transcriptomes data of fungi provided evidence of the existence of negative-sense RNA viruses in fungi before the first negative-sense RNA mycovirus was isolated [59].

In this study, we followed a metatranscriptomics approach to identify the mycovirus communities of six species of the genus *Alternaria*, which are associated with pear spot disease in China. We also identified near-full-length sequences of putative mycoviruses. We isolated a novel −ssRNA mycovirus from an *A*. *tenuissima* strain HB-15, which we designated as Alternaria tenuissima negative-stranded RNA virus 2 (AtNSRV2). We also characterized a novel +ssRNA mycovirus from an *A*. *tenuissima* strain SC-8, which we designated as Alternaria tenuissima deltaflexivirus 1 (AtDFV1). Through phylogenetic and sequence analyses, we found AtNSRV2 to be related to the viruses of the genus *Sclerotimonavirus* in the family *Mymonaviridae*. We also found that AtDFV1 is related to the family *Deltaflexivirus*. In this study, we identified viruses in *Alternaria* spp., for the first time, using total RNA sequencing. These results significantly expanded the number of *Alternaria* viruses and revealed the abundant diversity of the mycoviruses. 

## 2. Materials and Methods

### 2.1. Fungal Isolates and Culture Conditions

We selected 78 strains of *Alternaria* spp. from among isolated strains of pear black spot samples. We collected these samples from the primary among the 10 provinces that produce pears in China [20]. We used potato dextrose agar (PDA) plates to culture the strains in the dark at 28 °C. We used the sterile 25% glycerol solution to store the 5 mm mycelial agar discs at −80 °C. We used the partial region sequences of six loci to identify the strains: partial rDNA-ITS region, glyceraldehyde-3-phosphate dehydrogenase (GAPDH), translation elongation factor 1-alpha (TEF 1α), endo polygalacturonase (endoPG), Alternaria major allergen gene (Alt al), and histone 3 (His 3) [20]. Table 1 and Appendix A list the species of the strains used in this study and include the specific sources.

### 2.2. RNA Extraction and Sample Preparation for High-Throughput Sequencing

We prepared total RNA for high-throughput sRNA sequencing. We cultured 78 strains on cellophane membranes that overloaded PDA plates at 28 °C for 7 days. By mixing the mycelia in equal proportion, we ground the samples using liquid nitrogen. We used a TRIzol RNA extraction kit (Thermo Fisher, Waltham, MA, USA) to extract the total RNA, which we then clarified with chloroform in 2 mL tubes. We used ethanol precipitation to obtain the total nucleic acid fractions. We used 75% alcohol to wash the sample two times, and then dissolved them in water treated with diethylpyrocarbonate (DEPC). The total RNA was detected by agarose gel electrophoresis with 1.2% (*w*/*v*). Then, we determined the concentration and selected qualified samples for high-throughput sequencing.

### 2.3. High-Throughput Sequencing and Data Analysis

We sent the total RNA of the mixed strains to the Beijing Biomarker Technologies Company (Beijing, China), where it was constructed and sequenced for cDNA library. First, rRNA was removed, and then double-stranded cDNAs were synthesized using random hexamers (N6). The cDNA ends were repaired, and the A-tail was added and sequenced. The final library was obtained using the polymerase chain reaction (PCR). Then, the quality of the library was tested and sequenced using an Illumina HiSeq XTen platform.

A certain proportion of low-quality data is inevitable when obtaining raw data through sequencing. To ensure the reliability and accuracy of the analysis results, we had to preprocess the raw data, which were strictly controlled, and filtered as follows: we first removed the reads using adaptors, and then removed the low-quality reads for clean data. We analyzed the potential sequences of mycoviruses according to the splicing and assembly of clean data.

### 2.4. Validation of Virus-like Contigs by RT-PCR and Viral Sequencing Amplification

We isolated the total RNA of 78 strains and used reverse transcription polymerase chain reaction (RT-PCR) to investigate viruses in the fungal strains using specific primers, which were designed based on the assembled contigs (Appendix A). For cDNA synthesis, we added 1 μL random hexamers (N6), 2 μL DEPC-treated water, and 7 μL total RNA sample extracted from each tested strain to a 500 μL tube (RNase-free). After 8 min in boiling water bath, the sample was placed on ice for 3–5 min, the following system was added: 4 μL of DEPC-treated water, 4 μL of 5×M-MLV reverse transcription buffer, 1 μL of 2.5 mM dNTP, 0.5 μL of RNase inhibitor (TaKaRa, Dalian, China), 0.5 μL of M-MLV reverse transcriptase (PROMEGA, Madison, WI, USA). We mixed and centrifuged the sample, and then reverse transcribed the mixture for 1–2 h at 37 °C in a 20 µL reaction mixture. After reaction, obtained products were used for PCR or stored at −20 °C. The PCR products were electrophoresed in a 1.2% agarose gel, and stained with ethidium bromide (EB, 0.1 μg/Ml) for visualization on a UV trans-illuminator. We used a Pmd18-T vector (TaKaRa, Dalian, China) to ligate the purified PCR product, which was transformed into competent cells of *Escherichia coli* DH5α. We conduct the sequencing at Sangon Biotech Co. (Shanghai, China).

We used RT-PCR to obtain full-length sequences of the viruses. We determined virus sequences using primers designed from the data obtained from the assembled contigs (Appendix A). We also amplificated the cDNA ends sequence of virus using a SMARTer RACE 5′/3′ kit (TaKaRa, Dalian, China) according to the manufacturer’s instructions. We ligated the purified PCR product into a pMD18-T vector, which was transformed into *E*. *coli* DH5α. The sequencing was completed at Sangon Biotech Co. We determined three or more independent clones for each product in both orientations.

### 2.5. Sequences Alignment and Phylogenetic Analysis

We calculated sequence similarities using BLAST program on the NCBI database (https://www.ncbi.nlm.nih.gov (accessed on 22 October 2022)). According to the alignment information, the viruses related information (e.g., nucleic acid type, virus species, and the number of open reading frames [ORFs]) was preliminarily confirmed. We used DNAMAN (Lynnon Corporation, Pointe-Claire, Quebec, Canada) to conduct the sequence assembly. We used the ORF Finder website (http://www.ncbi.nlm.nih.gov/gorf/gorf.html (accessed on 22 October 2022)) to predict ORFs. We searched the CDD database (http://www.ncbi.nlm.nih.gov/Structure/cdd/wrpsb. cgi (accessed on 22 October 2022)) and the Pfam database (http://pfam.xfam.org/ (accessed on 22 October 2022)) to predict the conserved domains in sequences. We used MEGA 10 and applied the maximum likelihood method to construct the phylogenetic trees, which we tested with 1000 bootstrap replicates. We used MAFFT (http://www.ebi.ac.uk/Tools/msa/maft/ (accessed on 22 October 2022)) to conduct multiple sequence alignments of proteins encoded by the contigs and the reference mycovirus.

## 3. Results

### 3.1. Diversity of Alternaria Viruses

Reportedly, several phytopathogenic fungi have fungal viruses. We identified the presence of mycoviruses in the fungi of *Alternaria* and generated an RNA sequencing library with 78 strains (Table 1). The library was sequenced and assembled at a considerable depth. After removing rRNA from the assembled data, we obtained 67,011,714 reads by fragment reverse transcription, and screened out 66,709,434 clean reads by quality control data. Overall, a total of 131,203 contigs with a total length of 82,578,727 bp were yielded by transcriptional splicing, database alignment, and coding region prediction. The N50 and N90 values were 812 and 291, respectively. The results showed that the quality of the spliced sequences was good. We obtained 26 contigs that were derived from mycoviruses from this sequence (Table 2).

We subjected the contigs to BLAST analysis and assigned the contigs to 21 putative novel mycoviruses. These mycoviruses were characterized by eight different viral families, including *Botourmiaviridae*, *Botybirnaviridae*, *Chrysoviridae*, *Deltaflexiviridae*, *Hypoviridae*, *Partitiviridae*, *Mybuviridea*, *Mymonaviridae*, and *Narnaviridae*. Among them, 12 viruses belonged to +ssRNA viruses, 5 viruses belonged to dsRNA viruses, and 4 viruses belonged to −ssRNA viruses (Table 2).

### 3.2. Detection and Validation of Alternaria Viruses by RT-PCR

According to the sequence of contigs obtained by high-throughput sequencing, specific primers were designed for RT-PCR detection. The results showed that 21 viruses could be detected in 22 strains of *Alternaria* spp. (Figure 1), including 16 strains of *A*. *tenuissima*, 3 strains of *A*. *alternata*, 1 strain each of *A*. *arborescens*, *A*. *gossypina*, and *A*. *gaisen*. These results indicated that the putative viral sequences were reliable. Strain G-21-2 of *A*. *tenuissima* and KEL-4-4 of *A*. *arborescens* were infected by four viruses (Figure 1). Strain HB-15 was infected by three viruses. Strains G-9 and AH-25 of *A*. *tenuissima* were infected by two viruses (Figure 1). Others harbored only one virus.

### 3.3. Positive-Sense Single-Stranded RNA Virus

We used the obtained contigs and identified 12 positive-sense single-stranded RNA viruses. These sequences could be classified into four families, including *Hypoviridae*, *Narnaviridae*, *Deltaflexiviridae*, and *Botourmiaviridae* (Table 2).

The family *Hypoviridae* contains the genus *Hypovirus*. Contig 5 (14,170 nt) had a large ORF (484–13,302 nt), which encoded a 4272 aa protein. We found that this predicted amino acid sequence best resembled the RdRp of Alternaria alternata hypovirus 1 (AaHV1, GenBank: QFR36339) and had a 97.89% homology (Table 2). Thus, this virus should be a new strain of AaHV1. This hypovirus was detected in *A*. *tenuissima* strain JL-7 and G-9 (Figure 1). Therefore, the virus might be called Alternaria tenuissima hypovirus 1 (AtHV1).

The family *Mitoviridae* had only one genome and encoded one ORF, including genera *Duamitovirus*, *Kvaramitovirus*, *Triamitovirus,* and *Unuamitovirus.* The contig 5012 (2170 nt) had a large ORF (72–2048 nt) that encoded a 658 aa protein. We found that a predicted amino acid sequence of the protein best resembled the polyprotein of Alternaria arborescens mitovirus 1 (AaMV1, GenBank: YP_009270635), which had a 91.04% homology. Therefore, this virus was a strain of AaMV1 that belonged to the genus *Duamitovirus* in the family *Mitoviridae* (Table 2). We detected this virus in *A*. *tenuissima* strains G-9 and G-21-2 (Figure 1) and called it Alternaria tenuissima mitovirus 1 (AtMV1).

*The family Narnaviridae* had only one genome and encoded one ORF, including one genus *Narnavirus.* The contig 5919 (2004 nt) had a 98.04% homology with the RdRp of Neofusicoccum parvum narnavirus 2 (NpNV2, GenBank: QDB74995), which meant this virus was a strain of NpNV2 and belonged to the genus *Narnavirus*. The results of RT-PCR revealed that *A*. *tenuissima* strain AH-29 harbored this virus and was called Alternaria tenuissima narnavirus 1 (AtNV1) (Figure 1).

The twelve genera *Botoulivirus*, *Betabotoulivirus*, *Magoulivirus*, *Penoulivirus*, *Ourmiavirus*, *Rhizoulivirus*, *Betarhizoulivirus*, *Scleroulivirus*, *Betascleroulivirus*, *Deltascleroulivirus*, *Gammascleroulivirus,* and *Epsilonscleroulivirus* are in the family *Botourmiaviridae*. In this study, eight of the obtained contigs were homologous with numbers of this family, including two magouliviruses, one ourmiavirus, one botoulivirus, one betabotoulivirus, one betascleroulivirus, one deltascleroulivirus, and one unclassified (Table 2). According to the BLASTx result, contig 2423 (2972 nt) best resembled the RdRp of Cladosporium cladosporioides ourmia-like virus 2 (CcOLV2, GenBank: QDB75008) at 96.32% similarity (Table 2). We determined that this virus was a strain of CcOLV2, which was detected in *A*. *tenuissima* strains G-5, G-41, GS-8, and AH-25 (Figure 1). We called it Alternaria *tenuissima* ourmia-like virus 1 (AtOLV1). AtOLV1 was also detected in *A*. *arborescens* strain KEL-4-4 and *A*. *gossypina* strain SC-16 (Figure 1). We found the greatest similarity with contig 2672 (2845 nt), which best resembled the RdRp of Alternaria alternata magoulivirus 1 (AaMOV1, GenBank: UOV22670) at 98.90% homology and Plasmopara viticola associated ourmia-like virus 32 (DMG-F_40507PavOLV32, GenBank: QGY72562) at 98.07% homology (Table 2). Therefore, the virus was a strain of AaMOV1. We detected the virus in *A*. *tenuissima* strain GZ-1 and called it Alternaria tenuissima ourmia-like virus 2 (AtOLV2). According to the BLASTx result, contig 4360 (2324 nt) best resembled the RdRp of Plasmopara viticola associated ourmia-like virus 37 (DMG-B_Contig4PavOLV37, GenBank: QGY72567) at 94.75% similarity (Table 2). We determined that this virus was a strain of DMG-B_Contig4PavOLV37. We detected this virus in *A*. *tenuissima* strain SC-12 (Figure 1) and called it Alternaria tenuissima ourmia-like virus 3 (AtOLV3). According to the BLASTx result, contig 5365 (2102 nt) best resembled the RdRp of Plasmopara viticola associated ourmia-like virus 64 (DMG-A_13793PavOLV64, GenBank: QGY72594) at 50.00% similarity (Table 2). We detected this virus in *A*. *arborescens* strain KEL-4-4 and called it Alternaria arborescens ourmia-like virus 1 (AarOLV1). According to the BLASTx result, contig 2521 (2918 nt) best resembled the RdRp of Penicillium sumatrense ourmia-like virus 1 (PsOLV1, GenBank: QDB75000) at 86.38% similarity (Table 2). We determined that this virus was in *A*. *alternata* strain SC-32 (Figure 1) and called it Alternaria alternata ourmia-like 1 (AalOLV1). According to the BLASTx results, contig 6218 (1950 nt) best resembled the RdRp of Plasmopara viticola associated ourmia-like virus 52 (DMG-E_27866PvaOLV52, GenBank: QGY72582) at 64.17% similarity (Table 2). We detected this virus in *A*. *tenuissima* strain G-21-2 (Figure 1) and called it Alternaria tenuissima ourmia-like virus 4 (AtOLV4). Therefore, AtOLV4 was a novel ourmia-like virus. According to the BLASTx result, contig 3454 (2568 nt) best resembled the RdRp of Plasmopara viticola associated ourmia-like virus 65 (DMG-A_388322PavOLV65, GenBank: QGY72595) at 94.85% similarity (Table 2). Therefore, the virus was a strain of DMG-A_388322PavOLV65. We detected this virus in *A*. *tenuissima* strain AH-25 (Figure 1) and called it Alternaria *tenuissima* ourmia-like virus 5 (AtOLV5). AtOLV5 was also detected in *A*. *gaisen* strain AH-20 (Figure 1). According to the BLASTx results, contig 19,628 (946 nt) and contig 49,207 (482 nt) best resembled the RdRp of Colletotrichum fructicola ourmia-like virus 2 (CfOLV2, GenBank: UOV22974) at 89.04% and 86.44% similarity, respectively. We detected this virus in *A*. *tenuissima* strains SC-12 and HB-15 (Figure 1). Therefore, we called it Alternaria tenuissima ourmia-like virus 6 (AtOLV6) and AtOLV6 might be a novel virus. Unfortunately, not constructing phylogenetic tree for lacking the GDD motif in the amino acid sequence of RdRp of AtOLV6 encoded by contig 19,628.

We analyzed the relationships among three novel viruses and other mycoviruses in the family *Botourmiaviridae*. Then, we constructed a phylogenetic tree based on the RdRp amino acid sequence of AarOLV1, AalOLV1, AtOLV4, and other related viral sequences, such as ourmiaviruses, sclerouliviruses, magouliviruses, botouliviruses, and mitoviruses. However, this was not used in phylogenetic analysis due to the lack of GDD motif in the amino acid sequence of RdRp of AtOLV6 encoded by contig 19,628. In the obtained phylogenetic tree, AalOLV1 was grouped with some botoliviruses (Figure 2). AtOLV4 and some betabotoliviruses were grouped in a branch (Figure 2). AarOLV1 was clustered with some deltasclerouliviruses in a group (Figure 2). As a result, we identified AalOLV1 and AtOLV4 as the new members of the recent genus *Botolivirus* and *Betabotolivirus*, and AarOLV1 as a new member of the genus *Deltasclerouliviruses* within the family *Botourmiaviridae*.

The BLASTx results showed that contig 73 (8352 nt) was similar to RdRps of Agrostis stolonifera deltaflexivirus 1 (AsDFV1, 98.83% identity, GenBank: QQG34628), Alternaria alternata deltaflexivirus 1 (AaDFV1, 98.34% identity, GenBank: QTZ98076), Erysiphe necator associated deltaflexivirus 1 (EnDFV1, 95.61% identity, GenBank: QKN22722), and Triticum polonicum deltaflexivirus 1 (TpDFV1, 96.84% identity, GenBank: QQG34637). Contig 73 was identified from a *A*. *tenuissima* strain SC-8 by RT-PCR amplification. We designated this virus to be Alternaria tenuissima deltaflexivirus 1 (AtDFV1).

#### 3.3.1. Characterization of the Virus AtDFV1 Genome

We obtained the cDNA sequence of the AtDFV1 segment by combining the contig 73, RT-PCR, and RLM-RACE sequences. The full-length cDNAs of AtDFV1 were 8402 nt, excluding the ploy(A), which had a GC content of 51.4%. According to the BLASTn, this sequence best resembled segment Agrostis stolonifera deltaflexivirus 1, and had a 93.47% homology (GenBank: MW328744, E-value = 0.0, coverage 99%). We found that the AtDFV1 5′- and 3′-untranslated regions (UTR) were 19 nt and 146 nt long, respectively. The AtDFV1 contains four ORFs (I-IV). ORFs I and II were arranged in a line on the genome, and ORFs III and IV had 55 nt overlaps (Figure 3A).

We determined that ORF I (20–6241 nt) encoded a protein (P1) that had 2073 amino acid (aa) residues and a mass of 232 kDa. According to BLASTp, the P1 protein had 98.74% homology with the RdRp of Agrostis stolonifera deltaflexivirus 1 (AsDFV1) (GenBank: QQG34628, E-value = 0.0, coverage 99%). The P1 protein also had high homology with the RdRp of some deltaflexivirus (see Appendix A). In addition, four motifs domains were found from the P1 protein, including viral methyltransferase (Mtr, Pfam01660, position 191 to 511 aa, E-value = 1.1 × 10^−31^), viral RNA helicase (Hel, Pfam01443, position 1212 to 1478 aa, E-value = 0.082), RNA-dependent RNA polymerase (RdRp, Pfam00978, position 1781 to 1934, E-value = 2.9 × 10^−5^), and protein of unknown function (DUF3581, Pfam12119, position 807 to 838, E-value = 0.5). Therefore, ORF I encoded the viral RdRp with a methyltransferase and viral RNA helicase (Figure 3A). We observed that the homologous domains from selected other deltaflexivirus aligned with conserved domains (Mtr, Hel, and RdRp) of the putative P1 of AtDFV1. We used a pairwise comparison to identify the amino acid sequence identity (Figure 3B–D). ORF II (6622–6978 nt) encoded a protein (P2) with 118 aa residues and a mass of 13 kDa. According to BLASTp, the P2 protein was 98.31% similar to a hypothetical protein 2 (HP2) of AsDFV1 (GenBank: QQG34629, E-value = 0.0, coverage 100%). We found a motif that was homologous with viral RNAse III in mycoviruses (Mycovirus_RNAse, Pfam20614, position 28 to 99, E-value = 7.0 × 10^−10^). We also found that ORF III (7290–7823 nt) encoded a protein (P3) that had 177 aa residues and a mass of 18 kDa. According to BLASTp, the P3 protein was exactly the same as hypothetical protein 3 (HP3) of AsDFV1 (accession: QQG34630, E-value = 1.0 × 10^−112^, coverage 100%). We did not find any motifs in the P3 amino acid sequence. We observed that ORF IV (7780–8256 nt) encoded a protein (P4) with 158 aa residues and a mass of 16 kDa. According to BLASTp, the P4 protein was 97.47% similar to a hypothetical protein 4 (HP4) of AsDFV1 (accession: QQG34631, E-value = 6.0 × 10^−90^, coverage 100%). We did not find any motifs in the P4 amino acid sequence. We deposited the corresponding sequences in GenBank (accession: ON263576).

#### 3.3.2. Phylogenetic Analysis of AtDFV1

We constructed a maximum-likelihood phylogenetic tree using the entire replicase of AtDFV1 and members of *Alphaflexiviridae*, *Deltaflexiviridae*, and *Gammaflexiviridae*. We verified the existence of motifs I-VI in AtDFV1 and members of the deltaflexivirus family due to the amino acid alignment with the predicted RdRp (Figure 3D). According to this phylogenetic analysis, AtDFV1, AaDFV1, EnDFV1, and TpDFV1 were related to viruses in the family *Deltaflexiviridae* (Figure 4).

### 3.4. Double Strand RNA Virus

In this study, five mycoviruses belonged to dsRNA viruses, of which four viruses could be classified into the families *Chrysoviridae*, *Partitiviridae*, and the genus *Botybirnavirus*, and one had no classification status.

According to the BLASTx results, four contigs were homologous at 96.90–100.00% with the corresponding proteins encoded by Alternaria alternata chrysovirus 1 (AaCV1), which was a betachrysovirus in the family *Chrysoviridae* (Table 2). The BLASTx search also revealed that contig 1410 (3617 bp) had a 99.37% identity with the RdRp of AaCV1 (GenBank: QJW39304). Contig 2756 (2814 bp) shared a 100.00% similarity with the CP of AaCV1. Therefore, these contigs might belong to a new strain of the AaCV1. As shown in Figure 1, we detected the virus using the special primers based on contig 2756 in *A*. *tenuissima* strain SC-8 and called it as Alternaria tenuissima chrysovirus 1 (AtCV1).

At present, the family *Partitiviridae* contains five genera, namely, *Alphapartitivirus*, *Betapartitivirus*, *Cryspovirus*, *Deltapartitivirus,* and *Gammapartitivirus*. Contig 11,000 (1393 bp) had 99.05% homology with a coat protein of Alternaria dianthicola partitivirus 1 (AdPV1, GenBank: UYZ32457) (Table 2). Therefore, the virus corresponding to this sequence was a new strain of AdPV1. The RT-PCR results showed that this virus was detected in *A*. *tenuissima* strains G-21-2 and G-24-2. Therefore, we called it Alternaria tenuissima partitivirus 2 (AtPV2). Contig 7155 (1808 bp) shared 89.38% similarity with a dsRNA1 segment of Alternaria tenuissima partitivirus 1 (AtPV1, GenBank: MT648466). According to the BLASTx result, contig 7155 best resembled the RdRp of AtPV1 with 94.59% similarity (Table 2). Therefore, the virus corresponding to this sequence might be a new strain of AtPV1. Because we detected this contig in the *A*. *alternata* strain KEL-9-7, we temporarily called it as Alternaria alternata partitivirus 1 (AaPV1).

The genus *Botybirnavirus* presently includes 21 members. BLASTn searches revealed that two contigs were notable for regions with very strong similarity to viral cap-pol fusion protein gene and hypothetical protein gene of Botryosphaeria dothidea botybirnavirus 1 (BdBRV1). BLASTx searches showed that the contig 206 (6162 bp) sequence shared high similarity with the cap-pol fusion protein gene of BdBRV1 (97.92% identity, GenBank: AXP19719), Bipolaris maydis botybirnavirus 1 (BmBRV1; 97.87% identity, GenBank: YP_009551519), and Sclerotinia sclerotiorum botybirnavirus 4 (SsBRV4; 98.32% identity, GenBank: QUE49104). The BLASTx searches showed that contig 253 (5808 bp) best resembled the hypothetical protein of BdBRV1 (98.81% identity, GenBank: AXP19720), BmBRV1 (98.42% identity, GenBank: YP_009551518), and SsBRV4 (98.47% identity, GenBank: QUE49105). Thus, these contigs might represent a virus that was a new strain of BmBRV1. According to the RT-PCR results, this virus was detected in *A*. *tenuissima* strains G-21-2 and GZ-2. Therefore, we called it *Alternaria* tenuissima botybirnavirus 1 (AtBRV1).

The BLASTn results revealed that the contig 10,828 (1417 bp) and contig 13,903 (1210 bp) were notable for regions that had a very strong similarity to the sequences of Alternaria longipes dsRNA virus 1 (AlRV1; GenBank: KJ817371, 91; 88% and 90.58% identity). According to BLASTx, contig 10,828 best resembled the hypothetical protein of AlRV1 with 97.96% homology and contig 13,903 best resembled the RdRp of AlRV1 with 97.84% homology (Table 2). Therefore, the virus was a strain of AlRV1. This virus was detected in *A*. *arborescens* strain KEL-4-4 and called it Alternaria arborescens dsRNA virus 1 (AaRV1).

Therefore, we determined that dsRNA viruses in the pooled RNA-Seq sample were not new mycoviruses.

### 3.5. Negative-Sense Single-Stranded RNA Viruses

Based on their RdRp amino acid sequences, we identified four negative-stranded RNA viral sequences in our samples. We assigned these viruses to two taxonomical groups of negative-strand viruses: two were assigned to the family *Mymonaviridea*, and two were assigned to the family *Discoviridae* (Table 2).

The BLASTx searches revealed that contig 52 (8970 nt) shared 56.36% homology with RdRp of Cryphonectria parasitica sclerotimonavirus 1 (CpSMV1, GenBank: QMP84020). We used RT-PCR with specific primers to investigate the tested strains and found that strain HB-15 harbored contig 52 virus (Figure 1). Therefore, we called it Alternaria tenuissima negative-stranded RNA virus 2 (AtNSRV2). The complete genome of AtNSRV2 was obtained using RT-PCR and RACE. According to BLASTx, contig 15,899 (1079 nt) best resembled the putative nucleocapsid of Plasmopara viticola lesion associated mymonavirus 1 (PvLMMV1; GenBank: QHD64781, 43.12% identity) and ORF3 of Botrytis cinerea mymonavirus 1 (BcMMV1; GenBank: AXS76908, 29.13% identity). Thus, we determined this virus was a novel mymonavirus. As shown in Figure 1, we detected this virus in *A*. *tenuissima* strain G-21-1 and called it Alternaria tenuissima negative-stranded RNA virus 3 (AtNSRV3).

We found that contig 35 (10,267 nt) had one large ORF (<1–10,230 nt) that encoded a 3410 aa protein. According to the motif scan results, this protein contained a conservative Bunyavirus RNA-dependent RNA polymerase domain from 1886 aa to 2499 aa (pfam04196; E-value = 4.8 × 10^−11^). Furthermore, we detected a type II toxin-antitoxin system motif from amino acids 2812 to 2902 (E-value = 0.23) and a L protein N-terminus motif from 2812 aa to 2902 aa (E-value = 0.076) in the hypothetical protein. According to BLASTp, the protein had low homology with the RdRp of Botrytis cinerea negative stranded RNA virus 10 (BcNSRV10) with only 30.71% similarity, and Macrophomina phaseolina negative-stranded RNA virus 1 (MpNSRV1) had only 32.24% similarity. As shown in Figure 1, we detected this virus in *A*. *tenuissima* strain HB-15 and called this novel virus Alternaria tenuissima negative-stranded RNA virus 4 (AtNSRV4). Contig 192 (6528 nt) contained one large ORF (54–6473 nt) that encoded a 2139 aa protein. According to the motif scan results, this protein contained a conservative Bunyavirus RNA-dependent RNA polymerase domain from 545 aa to 1224 aa. In addition, we detected a Saccharopine dehydrogenase C-terminal domain from amino acids 1938 to 2012 (E-value = 0.85) in the hypothetical protein. According to BLASTp, this protein had low homology with the RdRp of Coniothyrium diplodiella negative-stranded RNA virus 1 (CdNSRV1) with 68.79% similarity. As shown in Figure 1, we detected this virus in *A*. *arborescens* strain KEL-4-4 and called the novel virus Alternaria arborescens negative-stranded RNA virus 1 (AaNSRV1).

#### 3.5.1. Characterization of the Virus AtNSRV2 Genome

We determined the complete cDNA sequence of AtNSRV2 using sequence verification and terminal cloning. The full-length sequence of AtNSRV2 was 9067 nt. The GC content of the whole genome was 53.3%, the 5′ and 3′-untranslated regions (UTRs) were 300 nt and 85 nt long, respectively. We predicted that the full length of the AtNSRV2 had five major ORFs (ORFs I-V). These non-overlapping ORFs were arranged in a line along the viral genome (Figure 5A). We also identified the conserved noncoding sequences (3′-AUUU/AAAUAAAACUUAGGA-5′), which were downstream the ORFs (Figure 5B). Because the gene-junction sequences were ubiquitous in the viral genome, we determined that they were a characteristic feature of the mononegaviruses. We deposited the nucleotide sequence of AtNSRV2 in GenBank (accession number: OP566533).

We found that ORF I encoded a protein with 253 amino acid residues and had a mass of 29 kDa. According to BLASTp, the protein had 33.62% homology with a hypothetical protein of Botrytis cinerea negative-stranded RNA virus 4 (BcNSRV4) (GenBank: QJW39407). We also obtained three motifs from the hypothetical protein encoded ORF I, including Family of unknown function (DUF5798; position 118 to 179 aa; pfam19111, E-value = 0.0032), LRRC37A/B like protein 1 C-terminal domain (LRRC37AB_C; position 198 to 251 aa; pfam14914, E-value = 0.073), and LMBR1-like membrane protein (LMBR1; position 100 to 230 aa; pfam04791, E-value = 0.098). We found that ORF II encoded a protein of 402 amino acid residues and had a mass of 44 kDa. According to BLASTp, it was 47.46% similar to the N protein (nucleoprotein) of the SsNSRV-1 (GenBank: YP_009094314). We discovered that ORF III encoded a protein of 52 amino acid residues and had a mass of 6 kDa. We did not find any significant similarity protein sequences in the BLASTp search. We found that ORF IV encoded the largest protein of 1931 amino acids in length and had a molecular mass of 220 kDa. According to BLASTp, the L protein was 56.48% similar to the RdRp of CpSMV1 (GenBank: QMP84020). The P4 protein of AtNSRV2 also had high homology with the RdRp of other −ssRNA mycoviruses (see Appendix A). The conserved domain predicted that the protein contained a mononegavirales mRNA-capping region V domain (Mononeg_mRNAcap; position 1127 to 1272 aa; pfam14318, E-value = 1 × 10^−6^) and a mononegavirales RdRp domain (Mononeg_RNA_pol; position 21 to 1041 aa; pfam00946, E-value = 3.3 × 10^−110^). Based on the multiple alignment of the sequences of the RdRp amino acid of AtNSRV2 and other related viruses from *Sclerotimonavirus*, we identified four conserved motifs (I–IV) (Figure 5C). We found that ORF V encoded a protein of 186 amino acid residues and had a mass of 20 kDa. According to BLASTp, the protein best resembled ORF4 of Plasmopara viticola lesion associated mononega virus 2 (PvLAMV2), with a 41.67% similarity (GenBank: QHD64788). The conserved domain predicted that the protein contained a Mannosidase Ig/CBM-like domain (Mannosidase_ig; position 61 to 102 aa; pfam17786, E-value = 0.16).

#### 3.5.2. Phylogenetic Analysis of the Novel −ssRNA Viruses

We constructed a phylogenetic tree based on the RdRp aa sequences of AtNSRV2, AaNSRV1, and AtNSRV4 as well as members of the families *Mymonaviridae* and *Discoviridae*. We found that AtNSRV2 formed a supported clade with the following viruses: Soybean leaf-associated negative-stranded RNA virus 1 (SlaNSRV-1), Fusarium gramineartm negative-stranded RNA virus 1 (FgNSRV1), AtNRV1, BcNSRV3, SlaNSRV2, and CpSMV1 (Figure 6). They were close to the clade and included members of the genus *Sclerotimonavirus* but were separate from other genera in the family *Mymonaviridae* (Figure 6). Therefore, according to genomic characteristics and phylogenetic analysis, we proposed that AtNSRV2 should be a new member of the genus *Sclerotimonavirus* in the family *Mymonaviridae*. We also identified novel −ssRNA mycoviruses that should be grouped with members in the family *Discoviridae*. We included AaNSRV1 in a group with CdNSRV1, BcNSRV2, and Fusarium poae negative-stranded virus 2 (FpNSRV2), which were classed a genus *Orthodiscovirus* in the family *Discoviridae*. We also included AtNSRV4 in a clade with the members in the family *Discoviridae*, but we considered it to be part of different groups. Therefore, AaNSRV1 and AtNSRV4 are in two highly supported groups inside the family *Discoviridae* (Figure 6).

The genomic −ssRNA of a novel mycovirus called AtNSRV2 was from strain HB-15. It was completely sequenced, and we found the virus to have a similar genome structure and high sequence similarity with members of the genus *Sclerotimonavirus* in the family *Mymonaviridae*.

## 4. Discussion

To search for novel mycoviruses, we used fungal strains from different regions to apply viral metagenomics to a mixed pool. This method allowed us to identify a great variety of new mycoviruses with different classes of genomes. In this study, we collected a total of 78 strains, including 58 *A. tenuissima*, 13 *A*. *alternata*, 3 *A*. *arborescens*, 2 *A*. *gaisen*, 1 *A*. *gossypina*, and 1 *A*. *longipes*, from pear spot disease from different regions of China, and mixed a fungal pool used to search for novel mycoviruses. As a result, we identified 21 putative viral sequences, most of which were nearly the full-length genome. We found 12 +ssRNA viruses, 5 dsRNA, and 4 −ssRNA viruses. We also identified sequences that were near full-length sequences in putative mycoviral genomes. Notably, the following eight distinct lineages were similar to these viral families: *Botourmiaviridae*, *Chrysoviridae*, *Deltaflexiviridae*, *Discoviridae*, *Hypoviridae*, *Partitiviridae*, *Mitoviridae*, *Mymonaviridae,* and *Narnaviridae*. In these putative viruses, nine were new mycoviruses. Through our analysis, we found that most of these viruses were dsRNA and ssRNA, and we did not find any DNA virus in the 78 strains. We used RT-PCR with specific primers, which had been designed based on the assembled contigs, and confirmed this variety of viruses. Two viruses from strains of *A*. *tenuissima* were provided a complete genome sequence using RT-PCR and RACE in our study.

Viruses are commonly present in *Alternaria* spp. Recently, 14 mycoviruses in 7 families have been associated with fungi from *Alternaria*. Of these fungi, we also identified AaMV1, AaHV1, AaCV1, and AlV1 in our study. These four viruses were founded from the different isolates of *A*. *arborescens*, *A*. *alternata*, and *A*. *longipes* [24,25,26,34,39,40]. In our study, 10 viruses were discerned initially from other hosts with more than 90% similarity were detected in *Alternaria* isolates. A larger body of research has found that mycovirus occurs in two fungi hosts that are taxonomically. Some dsRNA and ssRNA mycoviruses were previously identified as infecting other fungal species or fungal genera. For example, Bipolaris maydis botybirnavirus 1 (BmBBV1) was found in *Bipolaris maydis* and *Botryosphaeria dothidea* [60]. Ophiostoma novo-ulmi mitovirus 3a-Ld (OnuMV3a-Ld) was found in *Sclerotinia homoeocarpa* and *Ophiostoma novo-ulmi* [61], Previously, Helminthosporium victoriae virus 190S was identified in *Helminthosporium victoriae* and *Bipolaris maydis* [62,63], and the mitovirus Hymenoscyphus fraxineus mitovirus 1 (HfMV1) was found in the *Hymenoscyphus fraxineus* and *H*. *albidus* [64,65]. Another study idengified a mycovirus that infected different fungal strains through metatranscriptomics. Two viruses, Macrophomina phaseolina mitovirus 4 and Rhizoctonia solani mitovirus 10, were different strains of the same virus infected different hosts [59] and Botrytis fuckeliana totivirus 1 was found in *Botrytis cinerea* and *B*. *fuckeliana* [14]. Additionally, Sclerotinia sclerotiorum hypovirus 1-A, Sclerotinia sclerotiorum hypovirus 2, Sclerotinia sclerotiorum negativestranded RNA virus 5, Sclerotinia sclerotiorum partitivirus 2 were found in in *B*. *cinerea* and *S*. *sclerotiorum* [14]. Both *S*. *sclerotiorum* and *B*. *cinerea* are necrotrophic fungi. They have wide hosts ranges, and their genomes show high sequence identity as well as a similar arrangement of genes [66]. Given that these mycoviruses can be found in coinfections in common plant hosts, these mycoviruses may transfer horizontally between coinfecting fungi. An essential resource may be transmitted between different fungi hosts. For example, Sclerotinia sclerotiorum hypovirulence associated DNA virus 1 (SsHADV-1) could infect a mycophagous insect, *Lycoriella ingenua*, which could act as a transmission vector [67]. Cryphonectria hypovirus 1 (CHV1) could replicate and spread in *Nicotiana tabacum*, which is a model plant [68]. Plant-fungal-mediated routes may disseminate the same viruses between different fungi in nature.

We used RT-PCR and confirmed a variety of viruses in each strain. Our results identified 22 strains that were infected by various viruses. Interestingly, the *A*. *tenuissima* strain G-21-2 contained four viruses that belonged to four distinct lineages. Moreover, five strains showed coinfection by various viruses. Plant pathogenic fungal species often experience coinfection with multiple mycoviruses [69]. For example, an avirulent *R*. *solani* isolate DC-17 harbored 17 different mycovirus species that were assigned to eight or more families [70]. In addition, a single *Fusarium mangiferae* strain SP1 was coinfected by 11 mycoviruses that belonged to three families [71]. Sahin et al. detected eight new fungal viruses co-infecting a single isolate of the hypogeous ectomycorrhizal fungus *Picoa juniperi* using high-throughput sequencing [72]. Similar to isolate DC-17 and SP1, many plant pathogenic fungal species co-infected by different mycovirus, such as *B*. *dothidea* [73,74], *B*. *cinerea* [75], *Magnaporthe oryzae* [76], *R. necatrix* [77], *Rhizoctonia solani* [78], *Sphaeropsis sapinea* [79], *S*. *sclerotiorum* [80,81,82], and *S*. *nivalis* [83]. Therefore, a strain harbored different viruses might be for horizontal transmission of viruses.

Recently, deep sequencing has been used to identify the diversity of fungal viruses within fungal species from diverse geographic regions [52,55,70,71,82,84,85,86,87,88,89,90]. Unlike other dsRNA extraction and cloning methods, this approach could obtain viral information for different classified viruses regardless of their genome types [13,14,56,86]. Deep sequencing also has been used to identify mycoviruses from diverse fungal strains in an experimental project. For instance, using a high-throughput sequencing-based metatranscriptomic approach, 66 previously undescribed mycoviruses were obtained from five fungal species, including *Colletotrichum truncatum*, *Diaporthe longicolla*, *Macrophomina phaseolina*, *R*. *solani*, and *S*. *sclerotiorum* [13]. Multiple mycoviruses, which were coinfected in a fungal strain, were often efficiently identified by deep sequencing. For example, using deep sequencing, 17 different mycovirus that were assigned to different families were found in an *R*. *solani* isolate DC-17 [70]. Eleven mycoviruses were identified as being part of 3 families in a single *F*. *mangiferae* strain SP1[71]. Eight new fungal viruses that co-infected a single isolate of *P*. *juniperi* were found using high-throughput sequencing [72]. In addition, high-throughput sequencing has been used to detect mycoviruses on the phyllosphere and arbuscular mycorrhizal fungi in the roots [54,85]. For example, 22 putative mycovirus genomes have been organized into 10 taxonomic groups and assembled from soybean leaf metatranscriptomes [54]. The diversity, evolution, and annual variation of mycovirus in *S*. *sclerotiorum* within a single field for three years were investigated using the metatranscriptomic approach [86].

*Alternaria* spp. have been reported to have several RNA viruses, including dsRNA viruses, positive-sense ssRNA viruses, and other unidentified viruses [21,22,23,24,25,26,27,28,29,30,31,32,33,34,35,36,37,38,39,40,41,42,43,44,45]. The virus SsNSRV1 was the first member of the family *Mymonaviridae* [12]. Recently, two sclerotimonaviruses in this family have been found in *A*. *tenuissima* and *A*. *dianthicola*, respectively [44,45]. In this study, four negative-stranded RNA viruses have been identified. AtNSRV2 and AtNSRV3 were members of the genus *Sclerotimonavirus* in the family *Mymonaviridae*. The genome of the virus AtNSRV2 also was obtained. According to BLASTp, the L protein of AtNSRV2 was similar to the RdRp of SlaNRV2 with 55.85% identity. The virus SlaNRV2 was assembled from soybean leaf metatranscriptomes [54]. AaNSRV1 and AtNSRV4 were new members of the family *Discoviridae*. More research is needed to address these differences and to confirm their molecular and biological characterization. A lot of −ssRNA mycoviruses were classed into the order *Discoviridae* [3,4]. A recently reported BcNSRV1, RsNSRV4, and Macrophomina phaseolina negative-stranded RNA virus 1 (MpNSRV1) are new members of the order [8,13]. This is the first time that a mycovirus with a negative-stranded ssRNA genome has been reported to have infected an *Alternaria* strain.

## 5. Conclusions

In conclusion, diversity analysis of mycoviruses from 78 strains was executed by using high-throughput sequencing technology. The used stains were collected from 10 pear production areas and belonged to six species of the genus *Alternaria*. We excavated at least 21 different mycoviruses. The +ssRNA viruses belonged to the families *Mitoviridae*, *Narnaviridae*, *Deltaflexiviridae*, *Hypoviridae*, and *Botourmiaviridae*. The dsRNA viruses belonged to the families *Chrysoviridae*, *Partitiviridae*, and a genus *Botybirnavirus*, and one unclassified virus. The −ssRNA viruses belonged to the families *Mymonaviridae* of order *Mononegavirales*, and *Discoviridae* of order *Bunyavirales*. We also identified near-full-length sequences of mycoviral genomes that are putative. We isolated a novel −ssRNA mycovirus from an *A*. *tenuissima* strain HB-15, which we designated as AtNSRV2. We also characterized a novel +ssRNA mycovirus from an *A*. *tenuissima* strain SC-8, which we designated as AtDFV1. Through phylogenetic and sequence analyses, we found AtNSRV2 to be related to the viruses of the genus *Sclerotimonavirus* in the family *Mymonaviridae*. We also found that AtDFV1 is related to the family *Deltaflexivirus*. The results of this study significantly enhanced the number of *Alternaria* viruses and identified their abundant diversity.

## Figures and Tables

**Figure 1 viruses-14-02552-f001:**
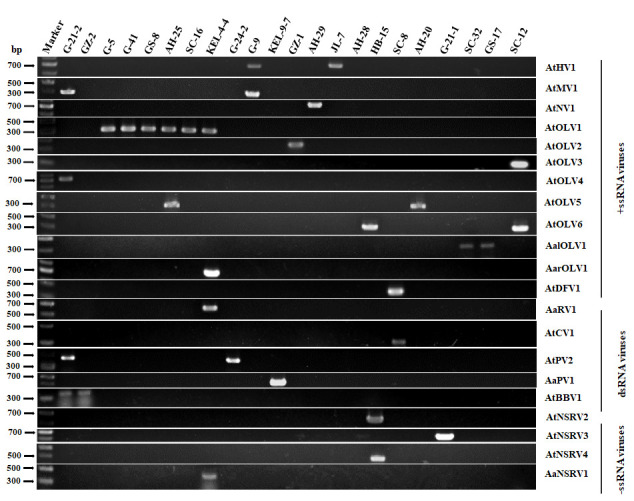
Detection of the 21 putative mycoviruses in different strains by RT-PCR. We used the assembled contigs to design the specific primers (see Appendix A). Only 22 strains of *Alternaria* species used for high-throughput sequencing were detected viruses. Lane Marker, DNA Marker II (TianGEN, Beijing, China). For abbreviations of virus names used in detection of the viruses, see Table 2.

**Figure 2 viruses-14-02552-f002:**
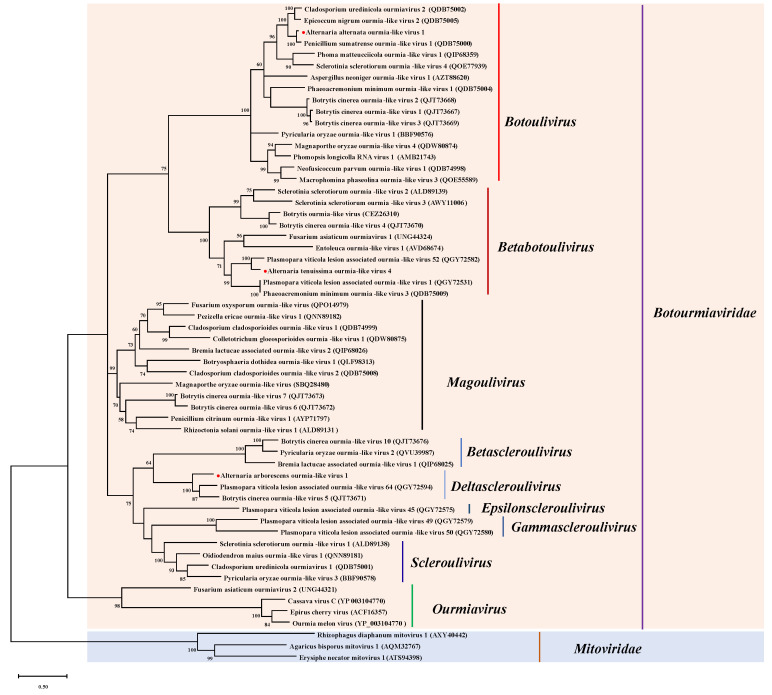
Phylogenetic analysis of AarOLV1, AalOLV1, AtOLV4. The novel viruses obtained by high throughput sequencing were indicated by a red circle (●). The data coverage percentages were shown by the numbers on the left of branches. We constructed a phylogenetic tree using the maximum likelihood method. We based the 1000 bootstrap replications on the best-fit protein evolution (LG+G+I+F) model. We set the gamma value at 2.

**Figure 3 viruses-14-02552-f003:**
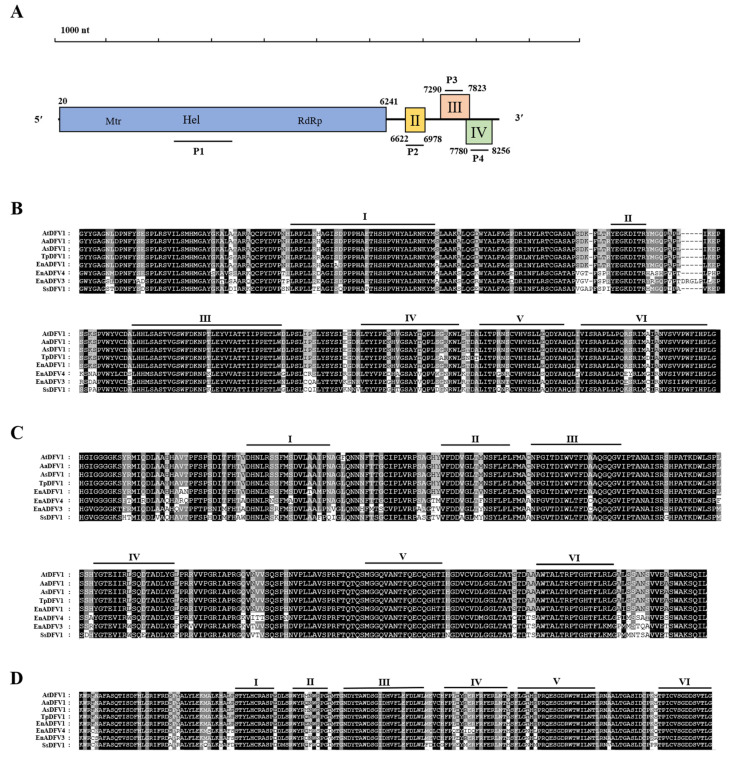
AtDFV1 genomic organization. (**A**) Organization and genome size with two conserved domains in the RdRp protein. Boxes indicate the position and size of each ORF and are labeled with Roman numerals, except for ORF I which encodes RdRp protein. (**B**–**D**) Amino acid sequences of the viral methyltransferase, viral RNA helicase, and RNA-dependent RNA polymerase, respectively, of AtDFV1 with members of the family of *Deltaflexivirus*. Black indicates the conserved sequences, shading indicates the conserved sequence level, and the darkest color indicates the most conserved sequence. For abbreviations of virus names and viral protein accession numbers used in alignment analysis, see Appendix A.

**Figure 4 viruses-14-02552-f004:**
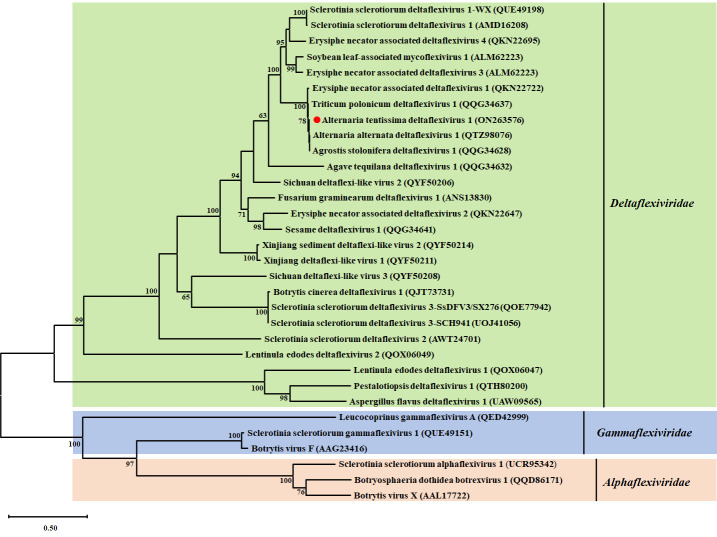
Phylogenetic analysis of AtDFV1 and other selected viruses using the maximum likelihood method with 1000 bootstrap replications. Comparisons of conserved the entire replication-associated polyprotein among members of several genera of the families *Alpha*-, *Gamma*-, and *Deltaflexiviridae*. Numbers on the left of branches indicate the data coverage percentage, and a red circle (●) indicates the position of virus AtDFV1. We used a best-fit model of protein evolution (LG+G+I) to construct the phylogenetic tree and set the gamma value to 2.

**Figure 5 viruses-14-02552-f005:**
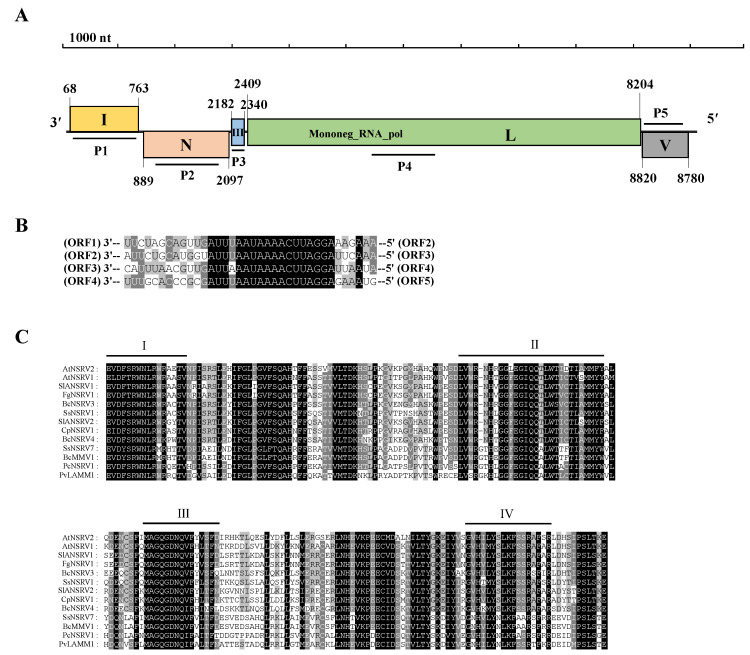
AtNSRV2 genomic organization. (**A**) Organization and genome size, and a conserved domain in the L protein. Boxes on the genome indicate the position and size of each ORF, which are labeled with Roman numerals, except for two ORFs, N and L, which encode the nucleoprotein (N) and RNA dependent RNA polymerase (RdRp) protein (L). (**B**) Putative gene junction regions among the ORFs in AtNSRV2. Alignment of the putative gene junction sequences is shown in a 3′-to-5′ orientation. (**C**) Amino acid sequence alignment of core RdRp motifs of AtNSRV2 and selected viruses from the genus *Sclerotimonavirus*. Conserved sequences are highlighted in black. Black highlights indicate the conserved sequences, shading indicates the conserved sequence levels, and the darkest color indicates the most conserved sequence. For abbreviations of virus names and viral protein accession numbers used in alignment analysis, see Appendix A.

**Figure 6 viruses-14-02552-f006:**
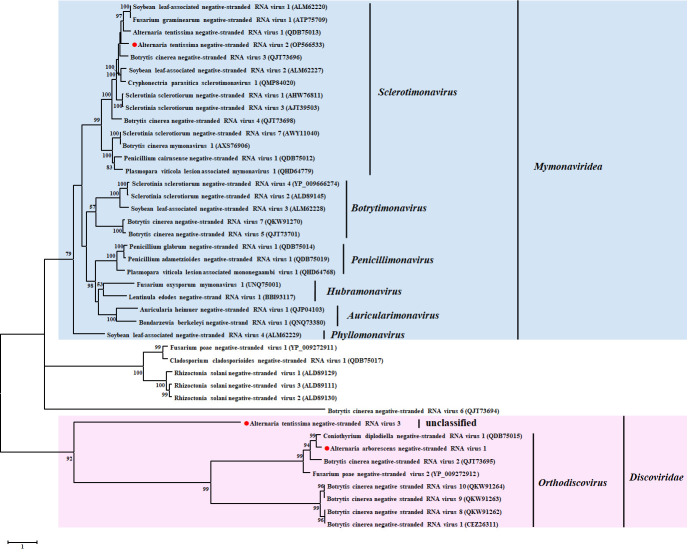
Phylogenetic analyses of the RdRp sequences including the novel −ssRNA mycoviruses and other selected viruses. We used the maximum likelihood method to conduct 1000 bootstrap replications. The data coverage percentage was shown by the numbers on the left of branches. The position of viruses AtNSRV2, AtNSRV4, and AaNSRV1 were shown by the red circle (●). We constructed the phylogenetic tree based on the best-fit protein evolution (LG+G+F) model and set the gamma value was 2.

**Table 1 viruses-14-02552-t001:** Origin of the strains of *Alternaria* species used for high-throughput sequencing in this study.

Area Source	Numbers of Strains	Species and Their Strains Numbers					
		*A. tenuissima*	*A. alternata*	*A. gossypina*	*A. arborescens*	*A. gaisen*	*A. longipes*
Anhui	8	5	2	0	0	1	0
Xinjiang	5	2	1	0	2	0	0
Shandong	6	5	1	0	0	0	0
Chongqing	8	6	1	0	0	1	0
Sichuan	9	5	1	1	1	0	1
Gansu	7	5	2	0	0	0	0
Yunnan	3	1	2	0	0	0	0
Jilin	2	2	0	0	0	0	0
Guizhou	2	2	0	0	0	0	0
Hubei	28	25	3	0	0	0	0
Total	78	58	13	1	3	2	1

**Table 2 viruses-14-02552-t002:** Best BLASTx matches of contigs obtained in this study.

No.	Contig Number	Contig Length (nt/bp)	Best Match	Host Strain	Name of Putative Virus	Protein	Cover %	aa Ident %	Taxon
+ssRNA Virus									
1	contig 5	14,170	Alternaria alternata hypovirus 1	*A*. *tenuissima*: G-9, JL-7	Alternaria tenuissima hypovirus 1 (AtHV1)	polyprotein	90	97.89	*Hypoviridae* *Hypovirus*
2	contig 5012	2170	Alternaria arborescens mitovirus 1	*A*. *tenuissima*: G-9, G-21-2	Alternaria tenuissima mitovirus 1 (AtMV1)	Polyprotein	94	91.04	*Mitoviridae Mitovirus*
3	contig 5919	2004	Neofusicoccum parvum narnavirus 2	*A*. *tenuissima*: AH-29	Alternaria tenuissima narnavirus 1 (AtNV1)	RdRp	91	98.04	*Narnaviridae Narnavirus*
4	contig 2423	2972	Cladosporium cladosporioides ourmia-like virus 2	*A*. *tenuissima*: G-5, G-41, GS-8, AH-25 *A*. *gossypina*: SC-16 *A*. *arborescens*: KEL-4-4	Alternaria tenuissima ourmia-like virus 1 (AtOLV1)	RdRp	60	96.32	*Botourmiaviridae* *Magoulivirus*
5	contig 2672	2845	Alternaria alternata magoulivirus 1	*A*. *tenuissima*: GZ-1	Alternaria tenuissima ourmia-like virus 2 (AtOLV2)	RdRp	76	98.90	
6	contig 4360	2324	Plasmopara viticola associated ourmia-like virus 37	*A*. *tenuissima*: SC-12	Alternaria tenuissima ourmia-like virus 3 (AtOLV3)	RdRp	81	94.75	*Botourmiaviridae* *Betascleroulivirus*
7	contig 5365	2102	Plasmopara viticola associated ourmia-like virus 64	*A*. *arborescens*: KEL-4-4	Alternaria arborescens ourmia-like virus 1 (AarOLV1)	RdRp	87	50.00	*Botourmiaviridae* *Deltascleroulivirus*
8	contig 2521	2918	Penicillium sumatrense ourmia-like virus 1	*A*. *alternata*: GS-17, SC-32	Alternaria alternata ourmia-like virus 1 (AalOLV1)	RdRp	79	86.38	*Botourmiaviridae* *Botoulivirus*
9	contig 6218	1950	Plasmopara viticola associated ourmia-like virus 52	*A*. *tenuissima*: G-21-2	Alternaria tenuissima ourmia-like virus 4 (AtOLV4)	RdRp	96	64.17	*Botourmiaviridae* *Betabotoulivirus*
10	contig 3454	2568	Plasmopara viticola associated ourmia-like virus 65	*A*. *tenuissima*: AH-25 *A*. *gaisen*: AH-20	Alternaria tenuissima ourmia-like virus 5 (AtOLV5)	RdRp	83	94.85	*Botourmiaviridae* *Ourmiavirus*
11	contig 19,628	946	Colletotrichum fructicola ourmia-like virus 2	*A. tenuissima*: SC-12, HB-15	Alternaria tenuissima ourmia-like virus 6 (AtOLV6)	RdRp	95	89.04	*Botourmiaviridae* unclassified
12	contig 73	8352	Agrostis stolonifera deltaflexivirus 1	*A*. *tenuissima*: SC-8	Alternaria tenuissima deltaflexivirus 1 (AtDFV1)	RdRp	73	98.83	*Deltaflexiviridae Deltaflexivirus*
**dsRNA virus**									
13	contig 10,828	1417	Alternaria longipes dsRNA virus 1	*A*. *arborescens*: KEL-4-4	Alternaria arborescens dsRNA virus 1 (AaRV1)	hypothetical protein	51	97.96	unclassified
	contig 13,903	1210				RdRp	91	97.84	
14	contig 1410	3617	Alternaria alternata chrysovirus 1	*A*. *tenuissima*: SC-8	Alternaria tenuissima chrysovirus 1 (AtCV1)	RdRp	92	99.37	*Chrysoviridae Betachrysovirus*
	contig 2756	2814				putative coat protein	82	100.00	
	contig 2896	2759				hypothetical protein	84	96.90	
	contig 7495	1762				hypothetical protein	78	100.00	
15	contig 11,000	1393	Alternaria dianthicola partitivirus 1	*A*. *tenuissima*: G-21-2, G-24-2	Alternaria tenuissima partitivirus 1 (AtPV2)	coat protein	90	99.05	*Partitiviridae*
16	contig 7155	1808	Alternaria tenuissima partitivirus 1	*A*. *alternata*: KEL-9-7	Alternaria alternata partitivirus 1 (AaPV1)	RdRp	85	94.56	*Partitiviridae* *Gammapartitivirus*
17	contig 206	6162	Botryosphaeria dothidea botybirnavirus 1	*A*. *tenuissima*: G-21-2, GZ-2	Alternaria tenuissima botybirnavirus 1 (AtBBV1)	cap-pol fusion protein	93	97.92	*Botybirnavirus*
	contig 253	5808				hypothetical protein	91	98.81	
**−** **ssRNA virus**									
18	contig 52	8970	Cryphonectria parasitica sclerotimonavirus 1	*A*. *tenuissima*: HB-15	Alternaria tenuissima negative-stranded RNA virus 2 (AtNSRV2)	RdRp	64	56.36	*Mymonaviridea* *Sclerotimonavirus*
19	contig 15,899	1079	Plasmopara viticola lesion associated mymonavirus 1	*A*. *tenuissima*: G-21-1	Alternaria tenuissima negative-stranded RNA virus 3 (AtNSRV3)	nucleocapsid	59	43.12	
20	contig 35	10,267	Botrytis cinerea negative stranded RNA virus 10	*A*. *tenuissima*: HB-15	Alternaria tenuissima negative-stranded RNA virus 2 (AtNSRV4)	RdRp	49	30.71	*Discoviridae* *Orthodiscovirus*
21	contig 192	6528	Coniothyrium diplodiella negative-stranded RNA virus 1	*A*. *arborescens*: KEL-4-4	Alternaria arborescens negative-stranded RNA virus 1 (AaNSRV1)	RdRp	97	68.76	*Discoviridae* unclassified

## Data Availability

The sequences reported in the present manuscript have been deposited in the GenBank database under accession numbers ON263576 and OP566533.

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
