# Peer review of "Novel Mycoviruses Discovered from a Metatranscriptomics Survey of the Phytopathogenic Alternaria Fungus"

_viruses, 2022, doi:10.3390/v14112552_

Round 1
Reviewer 1 Report
In the present study, authors identified the mycovirus communities from 78 strains in 6 species of the genus Alternaria sampled from 10 pear production areas using high throughput sequencing technology. The findings expand the number of Alternaria viruses and reveal the diversity of the mycoviruses. But some revisions are needed.
Major points:
1. The manuscript including Introduction, Results , Discussion and Conclusion need to be further refined and condensed;
2. There is opportunity for some corrections in English style and grammar here and there throughout the manuscript, although generally understandable.
Minor remarks:
1. Line 133 change “The we” as “We”;
2. Line 141 change “The we” as “We”;
3. Line 141 delete “we”;
4. Line 159 delete “we”;
5. Line 161-162 please rewrite the sentence “ we used staining with ethidium bromide (EB) for visualization.”
6. Line 166 delete “we”;
7. Line 198-199 please rewrite the sentence”We obtained 54 contigs derived that were derived from mycoviruses from this sequence. of all the contigs were listed in Table 2 gives the contig equences.”
Reviewer 2 Report
The paper Wang et al. reveals the mycoviral diversity of virome of the phytopathogenic fungus Alternaria through deep sequencing analysis. This analysis expands our knowledge of the viral species associated with Alternaria species. This information is important in the expansion of mycoviral database and will be useful in taxonomic organization/reorganization. However, several issues need to be addressed before its publication. I would advise authors to be cautious with how they write virus or taxon names. The paper requires a thorough language review.
Comment 1: I recommend the authors to reorganize the Table 1 to include fungal strains’ names and respective places of origin.
Comment 2: I find Table 2 confusing. There are several contigs all assigned to a single virus. For example, Mitovirus is mono-segmented and AtMV1 is hereby shown to have 5 contigs with a high percentage identity to Alternaria arborescens mitovirus 1. I fail to understand whether this amounts to variants or failure by the author to properly assemble the contigs.
Partitiviruses are bi-segmented and the RNAs encode for RdRP and CP separately. Table 2 reports a partitivirus with two CP encoding contigs. Another contig (No.16 ) encoded RdRP and lacks Cp encoding contig sequence. I may be tempted (I can be wrong) to assume that No.15 and 16 contigs may represent a single virus. However, there are also cases where it’s possible that low accumulated viral segments might filter out while analysing NGS data. But, based on the incomplete data analysis, it’s premature to make conclusions. Please refer to ICTV taxonomy whenever reporting novel species and strains.
Therefore, I would suggest that the authors reassemble these contigs and align them to get the complete viral sequences. Finally, incorporate the final result into the table. This will decrease the contig numbers in Table 2. In case contigs do not align, I recommend authors include this required information in the manuscript. Please consider this comment for other viruses as well such as Scleroulivirus (No. 7), Betachrysovirus (No.14) and Botybirnavirus (No. 17).
Comment 3: In Figure 1, RT- PCR amplicon for AaNSRV1, AalOLV1, AtOLV4 viruses (from A. arborescens, A. alternata and A. tenuissima respectively) are hardly visible. Better to be replaced with a good picture with a clear viral associated band. In addition, marker for AtMV1, AtOLV3, AgoOLV1, AgaOLV1, AtoLV1, AalOLV1, AtOLV3 AtOLV4, AtNSRV2, AtSNRV4 gel picture are not clearly visible. Also, please mention the type of marker used, 1Kb? And company sourced from.
Comment 4: The authors identified RNA viruses from their collections. To improve the paper's quality, the authors should include the dsRNA profile of some representative viruses. In addition, I would request that they can include some representative fungal pictures to that effect.
Comment 5: Figure 2 phylogenetic tree should be replaced with a high-quality picture.
The other concerns are listed below.
Abstract
Line 29: genus not genes
Introduction
Line 39: “fungi viruses” should be fungal viruses
Line 51: the sentence structure is not clear
Line 53: should be “alternata”
Line 58: Mitoviruses have been reclassified under Mitoviridae
Line 76: should be “numbers”
Line 81: should be “experience”
Line 82: should be mycovirus infection
Line 96: change viroomics to viromics
Line 97: Sentence not clear
Methods and Materials
Line 126: delete “for” before total RNA
Line 147: should be “analyzed”
Line 158: should be “mixed” not mixied
Line 160: add “for” before PCR
Results
Table 2: No. 2, taxon column, change “Narnaviridae” to “Mitoviridae”
Table 2: No. 11, host strain column, please confirm if it's A. tenussima “BH-15” or “HB-15”.
Line 330: add “respectively” after RNA-dependent RNA polymerase
Line 366: Figure 3 “C” or “D”? please confirm
Line 411: “(bp)” number of nucleotide sequence missing
Line 587: “Modern” or “Model”
